# Perivascular Adipose Tissue-Enhanced Vasodilation in Metabolic Syndrome Rats by Apelin and *N*-Acetyl–l-Cysteine-Sensitive Factor(s)

**DOI:** 10.3390/ijms20010106

**Published:** 2018-12-28

**Authors:** Satomi Kagota, Kana Maruyama-Fumoto, Saki Iwata, Miho Shimari, Shiori Koyanagi, Yayoi Shiokawa, John J. McGuire, Kazumasa Shinozuka

**Affiliations:** 1Department of Pharmacology, School of Pharmacy and Pharmaceutical Sciences, Mukogawa Women’s University, Nishinomiya 663 8184, Japan; k_maru@mukogawa-u.ac.jp (K.M.-F.); 22062@mwu.jp (S.I.); mw254182@mukogawa-u.ac.jp (M.S.); momotatu26@yahoo.co.jp (S.K.); yayoi_38@mukogawa-u.ac.jp (Y.S.); kazumasa@mukogawa-u.ac.jp (K.S.); 2Department of Medical Biophysics, Schulich School of Medicine & Dentistry, Western University, London, ON N6A 5C1, Canada; john.mcGuire@schulich.uwo.ca

**Keywords:** adipokine, mesenteric artery, metabolic syndrome, perivascular adipose tissue, vasodilation

## Abstract

Perivascular adipose tissue (PVAT) can regulate vascular tone. In mesenteric arteries of SHRSP.Z-*Lepr^fa^*/IzmDmcr rats (SHRSP.ZF) with metabolic syndrome, vascular dysfunction is compensated by PVAT-dependent mechanisms that disappear with increasing age. In this study, we investigated the mechanisms of the age-related changes and responsible factor(s) involved in the enhancing effects of mesenteric arterial PVAT in SHRSP.ZF. Acetylcholine- and sodium nitroprusside-induced relaxations of isolated arteries were greater with PVAT than without PVAT at 17 and 20 weeks of age (wks), and as expected, this enhancement by the presence of PVAT disappeared at 23 wks. PVAT mRNA levels of angiotensin II type 1 (AT1) receptor-associated protein was less and AT1 receptor was unchanged at 23 wks when compared to 20 wks. At 20 wks, the enhanced acetylcholine-induced relaxation by the presence of PVAT was inhibited by *N*-acetyl-l-cysteine (NAC). Acetylcholine-induced relaxation of arteries without PVAT was increased in the presence of exogenously added apelin. PVAT mRNA level of apelin was higher in SHRSP.ZF than in control Wistar-Kyoto rats, and the level was decreased with aging. These results suggest that AT1 receptor activation in PVAT, and changes in the regulation of apelin and a NAC-sensitive factor are related to the age-dependent deterioration of the vasodilation enhancing effects of mesenteric arterial PVAT in SHRSP.ZF.

## 1. Introduction

The perivascular adipose tissue (PVAT), located on the outside of blood vessels, has been recognized to regulate vascular tone and wall remodeling via the release of several bioactive substances, adipokines, resulting in the development of cardiovascular disease due to metabolic syndrome (MetS) [1,2]. It is important to note that MetS is a chronic condition with worsening symptoms (if untreated) with age. Therefore, we have assessed the role of PVAT in vascular dysfunctions using SHRSP.Z-*Lepr^fa^*/IzmDmcr rats (SHRSP.ZF), which were established as an animal model of MetS by crossing stroke-prone spontaneously hypertensive rats of the Izumo strain (SHRSP/Izm) to Zucker-fatty rats [3]. We proposed that PVAT plays a role in retaining vascular tonus under the conditions of vascular dysfunctions in response to nitric oxide (NO), and this compensatory system of PVAT will breakdown later in the time-course of MetS [4]. However, the identities of the PVAT-dependent factor(s) that enhance vasodilations and regulate the development and/or breakdown of the compensatory effects of PVAT in SHRSP.ZF with MetS are undetermined.

In the context of MetS, angiotensin II (Ang II) is considered a key regulator of not only vasomotor functions, but also inflammation of adipose tissue [2,5,6], and these activities are mediated mainly by activation of the Ang II type 1 (AT1) receptor. Previously, we demonstrated that AT1 receptor contributed to the development of vasomotor dysfunctions in the MetS rats [7,8]. Ang II signaling via AT1 receptor is negatively modulated by AT1 receptor-associated protein (ATRAP) [9,10]. ATRAP mRNA levels in adipose tissues from patients and mice with MetS are decreased compared to controls despite unchanged AT1 mRNA levels [11]. Specific enhancement of ATRAP in adipocytes inhibits the development of diet-induced obesity and adipose inflammation [12]. In the current study, we proposed to test whether ATRAP, and AT1 mRNA expressions in PVAT are altered with aging in SHRSP.ZF.

Apelin, vaspin, and visfatin are adipokines that are produced at higher levels in obesity and can act as diffusible paracrine vasorelaxing factors for regulation of vascular tone [13,14,15,16,17,18,19]. However, omentin is an adipokine that is produced at lower levels in obesity [20] and induces endothelium-dependent and independent vasorelaxation [21]. PVAT can generate hydrogen sulfide (H_2_S) which can also be considered an adipokine [22]. Vasodilation induced by a H_2_S donor was observed in samples of human mesenteric artery and this vasodilation was reduced by treatment with an inhibitor of NO synthase, l-NAME, or endothelium removal [23]. Interestingly, the enhancing effects by PVAT of SHRSP.ZF mesenteric arteries are inhibited by treatment with l-NAME or removing the endothelium [4]. It has been proposed that a reaction product of a chemical reaction between H_2_S and *S*-nitrosoglutathione, a so-called NO donor, putatively generates nitroxyl anions, likely in the more stable HNO form, and is involved in the relaxation of isolated rat thoracic aortas and mesenteric artery [24]. Indeed Angeli’s salt (sodium trioxodinitrate), which decomposes to HNO, causes vasodilation of the aorta [25,26], coronary artery [27], and mesenteric artery [28,29] in rodents; nitroxyl anion mediated relaxation was inhibited by *N*-acetyl-l-cysteine (NAC) in these studies. In the current study, we hypothesized that HNO, generated from NO and H_2_S, contributes to the enhancing effects of mesenteric arterial PVAT on vasodilation in the MetS rats.

Therefore, to assess the mechanisms underlying the disappearance of PVAT functions, we investigated whether mRNA transcripts of AT1 receptor and ATRAP in mesenteric arterial PVAT are changed with increasing age in SHRSP.ZF. To identify responsible factor(s) involved in the enhancing effects of mesenteric arterial PVAT, we tested whether vasorelaxation is enhanced in the presence of apelin, visfatin, vaspin, and omentin in mesenteric artery without PVAT of SHRSP.ZF at 20 weeks of age (wks), as seen in arteries with PVAT. Furthermore, as an initial test of the nitroxyl hypothesis, we assessed whether NAC inhibited the PVAT enhancing effects in SHRSP.ZF mesenteric artery. Based on the results of our study, we provide evidence to identify important factor(s) responsible for the compensatory effects of mesenteric arterial PVAT under the condition of vascular dysfunctions in MetS.

## 2. Results

### 2.1. Changes in PVAT Effects on Relaxation in Superior Mesenteric Arteries from SHRSP.ZF with Aging

First we looked at differences in the metabolic parameters and mesenteric arterial PVAT functions that occurred with aging, using SHRSP.ZF at 17, 20, and 23 wks, a narrower range of age than we have reported before. As shown in Table 1, body weight increased with aging, but waist length/body length, an index of abdominal obesity, was unchanged. Systolic blood pressure was significantly higher at 23 wks than that at 20 wks. There were no significant differences in serum triglycerides and glucose, while serum levels of insulin and TBARS, an index of oxidative stress, were higher at 23 wks than that at 20 wks. These data suggest that metabolic parameters became dramatically worse between 20 and 23 wks.

Next we compared the effects of mesenteric arterial PVAT from SHRSP.ZF on vasodilation in response to acetylcholine. Figure 1 and Figure 2 show acetylcholine-induced and sodium nitroprusside-induced vasodilations in mesenteric arteries with and without PVAT from SHRSP.ZF. The acetylcholine-induced relaxations were enhanced by the presence of PVAT at 17 and 20 wks. However, there were no significant differences in the relaxation between the presence and absence of PVAT at 23 wks. Similarly, sodium nitroprusside-induced relaxations were enhanced by the presence of PVAT at 17 and 20 wks, but were not altered at 23 wks. These results were consistent with our previous observations that PVAT enhanced vasodilations in response to not only endogenous NO, but also exogenous NO in mesenteric artery of SHRSP.ZF [4]. Interestingly, the time-course for disappearance of PVAT functions followed the time-course of worsening metabolic parameters described above.

### 2.2. Changes in mRNA Levels of AT1 Receptor, ATRAP, and Angiotensinogen in Mesenteric Arterial PVAT of SHRSP.ZF with Aging

Figure 3 shows mRNA levels in mesenteric arterial PVAT from SHRSP.ZF at 17, 20, and 23 wks. Level of AT1 receptor mRNA was unchanged in SHRSP.ZF with aging. In contrast, ATRAP mRNA was lower at 23 wks than at 20 wks. Furthermore, mRNA level of angiotensinogen was also decreased in PVAT at 23 wks compared to PVAT at 20 wks.

On the other hand, when mRNA levels in mesenteric arterial PVAT were compared between SHRSP.ZF and their normal control Wistar-Kyoto rats (WKY) at 20 wks, AT1 receptor level was approximately 2.6-times higher in SHRSP.ZF than that WKY, but there was no significant difference in ATRAP level between both strains (Figure 4A,B). Angiotensinogen level was approximately 0.7 times lower in SHRSP.ZF than WKY (Figure 4C).

### 2.3. Effects of Adipokines on the Relaxation in SHRSP.ZF Mesenteric Arteries without PVAT

In mesenteric arteries without PVAT [PVAT(−)] of SHRSP.ZF at 20 wks, omentin treatment did not significantly affect acetylcholine-induced relaxations, and its protein level in PVAT was undetectable in not only age-matched normal control WKY, but also SHRSP.ZF (Figure 5A and Figure 6A,E). Additionally, vaspin treatment did not affect the relaxation, but the protein level in PVAT was approximately 2.4 times higher (ratio of vaspin to β-actin, 0.360 ± 0.088 vs. 0.147 ± 0.051, *p* < 0.05) in SHRSP.ZF than that in WKY (Figure 5B and Figure 6B,F). In contrast, visfatin treatment increased acetylcholine-induced relaxation, but the protein level was approximately 0.7 times lower (ratio of visfatin to β-actin, 0.645 ± 0.047 vs. 0.897 ± 0.146, *p* = 0.176) in SHRSP.ZF than that in WKY (Figure 5C and Figure 6C,G). In the case of apelin, exposure of arteries to apelin significantly increased the relaxations, but the apelin protein in PVAT was undetectable in both strains (Figure 5D and Figure 6D,H). However, level of apelin mRNA in PVAT was approximately 2.5-times higher in SHRSP.ZF than that in WKY at 20 wks (Figure 7A). In contrast, there was no significant difference in the level between 17 and 20 wks, but it was lower at 23 wks in SHRSP.ZF (Figure 7B). The range of CT values for apelin in PVAT was 23.24–25.75.

### 2.4. Effects of NAC on the Relaxation in SHRSP.ZF Mesenteric Arteries with PVAT

HNO has been recognized to produce NO with H_2_S in arteries of rodents [24], and HNO donor induces relaxation in rat small mesenteric artery [29]. Therefore, as an initial step to investigate the possible presence of HNO in SHRSP.ZF mesenteric artery with PVAT, we tested whether NAC inhibited acetylcholine-induced relaxations in mesenteric arteries with PVAT [PVAT(+)] of SHRSP.ZF at 20 wks. As shown in the Figure 8 and Figure 9, treatment with NAC significantly decreased the relaxations.

## 3. Discussion

MetS is a chronic condition which worsens with time and thus, requires long-term strategies for clinical treatment. Our previous studies show that mesenteric PVAT has compensatory effects to maintain vascular tonus under the condition of impaired NO-mediated relaxations in SHRSP.ZF (at 20 wks), i.e., PVAT enhances vasodilation in response to either endothelium-derived NO induced by acetylcholine or exogenous NO derived from sodium nitroprusside in mesenteric artery of SHRSP.ZF [4,30,31]. However, the compensatory function of PVAT was impaired at a later stage of MetS (at 30 wks). The present study demonstrates that the disappearance of PVAT-mediated regulation of vasorelaxation in response to acetylcholine and sodium nitroprusside coincides with a decreased ATRAP mRNA level in adipose tissue of superior mesenteric arteries of SHRSP.ZF with MetS. These results suggest that AT1 receptor activity in PVAT is related to the deterioration of the enhancing effects of PVAT on vascular relaxations. Since the breakdown of the PVAT enhancing effect is observed later in the time-course of MetS (at 30 wks) [4], PVAT dysfunction, probably, occurs around 23 wks in the aging process in SHRSP.ZF with MetS. On the other hand, apelin mRNA in mesenteric PVAT was higher in SHRSP.ZF than controls and the incubation of tissues with exogenous apelin resulted in enhanced acetylcholine-induced relaxations. Its mRNA was decreased at a time coincident with impaired the enhancing effects of PVAT on vasodilation. Furthermore, treatment of tissues with NAC reduced relaxations in mesenteric artery with PVAT. Together these findings provide evidence that apelin and an NAC-sensitive factor, possibly HNO, are candidates for being the factors/molecules responsible for the enhancing effects of PVAT on vasodilation, which compensates for the impaired vasodilation observed in superior mesenteric artery of SHRSP.ZF with MetS.

In our current study, the metabolic abnormalities such as blood pressure and oxidative stress, worsened and PVAT’s compensatory effects disappeared at 23 wks in SHRSP.ZF. In arteries, the activation of AT1 receptor is known to be associated with the deterioration of the vascular functions in SHRSP.ZF [7,8]. However, when AT1 receptor mRNA in adipose tissue of SHRSP.ZF was examined, we found it was unchanged with aging. When considering the differences in its expression between MetS animals and age-matched normal control WKY, AT1 receptor level appears to be continuously higher in PVAT of SHRSP.ZF. In contrast, mRNA for ATRAP, which is a negative regulator of AT1 receptor, did decrease with a time-course matching the decline in PVAT function. We propose that activation of AT1 receptor in PVAT initiates the deterioration of the PVAT’s modification of vascular tone in MetS and declining ATRAP sustains the dysfunction. This mechanism is consistent with recent reports by another group that found ATRAP deficiency causes deterioration of metabolic disorders along with adipose inflammation, while increasing ATRAP activity ameliorated the abnormalities [11,12]. Furthermore, treatment with AT1 blocker, irbesartan, preserves the ratio of ATRAP to AT1 receptor in adipose tissue in MetS mice, leading to improvement of adipose tissue inflammation [32]. In contrast, losartan could not improve impaired PVAT’s modifications of acetylcholine-induced relaxations in aortas of fructose-fed rats [33]. ATRAP mRNA expression in adipose tissue is decreased significantly in patients with hypertension, but not in patients with diabetes [11]. These results may suggest that high blood pressure may have an important meaning for alteration of ATRAP levels in adipose tissue, leading to development of deterioration of PVAT effects in MetS. Further research is needed to investigate whether AT1 receptor antagonists can arrest the declining PVAT compensatory effects on vasomotor functions in mesenteric artery of SHRSP.ZF with MetS.

In our study, we looked at several adipokine peptides as being possible factor(s) involved in PVAT’s compensatory effects on vasomotor functions in mesenteric artery of SHRSP.ZF. Incubation of tissues with exogenous visfatin or apelin enhanced the acetylcholine-induced relaxation in mesenteric artery of SHRSP.ZF whereas omentin and vaspin did not enhance it. The content of visfatin in adipose tissue of SHRSP.ZF was less than in WKY. Therefore, it is unlikely that visfatin, omentin and vaspin are contributing to the PVAT mechanisms here. Our study’s findings are like those of another that reported apelin reversed an altered ACh-induced relaxation of thoracic aortas in diabetic *db*/*db* mice [34]. In addition, a higher level of apelin mRNA was observed in adipose tissue of SHRSP.ZF than that of WKY, while the level was decreased at older ages when the enhancing effects of PVAT disappeared in SHRSP.ZF. These findings suggest that apelin is involved in the enhancing effect of PVAT in MetS rat mesenteric artery, and changes in the level is associated with deterioration of the PVAT function. Although apelin mRNA was detected in PVAT of mesenteric artery of SHRSP.ZF, its protein content was below detectable limits by the Western blot method. This may raise the possibility that sources of apelin are not only the local adipose tissue around the artery but also that is synthesized by adipose tissues elsewhere as the circulating paracrine factor.

On the other hand, non-peptide signaling molecules such as H_2_S signaling could be a compensatory mechanism that maintains vascular tone, despite endothelial dysfunction, as described in short-term experimental obesity [35]. A reaction product of H_2_S and NO-donor, *S*-nitrosoglutathione, is proposed to regulate arterial tone in intrarenal arteries of patients with arterial hypertension [36]. We found that treatment with NAC reduced the enhanced acetylcholine-induced relaxation in mesenteric artery induced by the presence of PVAT in SHRSP.ZF at 20 wks. These findings provide evidence supporting the idea that HNO, which is formed by interaction of H_2_S and NO in vascular beds, may play a role in the compensatory effects of PVAT under vascular dysfunctions in the MetS rats. However, endogenous H_2_S production progressively decreases with increasing glycosuria values in non-obese diabetic mice [37] and a high-fat diet impairs endogenous H_2_S production by rat periaortic adipose tissue [38]. Interestingly, lipophilic atorvastatin, but not hydrophilic pravastatin, increases H_2_S production in PVAT of Wistar rat aortas [39]. Since the compensatory effects of PVAT disappears with aging in SHRSP.ZF with MetS [4], drugs that restore PVAT function, and thereby protect against development of cardiovascular diseases related to MetS, could have significant therapeutic potential.

Further study of the SHRSP.ZF model is needed to determine whether H_2_S and NO are produced from PVAT and vascular endothelium, respectively. In fact, H_2_S can be produced not only by adipose tissue, but also by vascular endothelium [40] and smooth muscle cells [41], and NO is released from adipose tissue and the endothelium [42]. Albeit technically challenging, direct measurements of these molecules under conditions of MetS would refine our interpretations. Additional work ought to determine whether H_2_S and NO or H_2_S alone suffices to mediate the PVAT functions in SHRSP.ZF. Furthermore, PVAT contains different phenotypes of adipocytes i.e., white, brown, and beige adipocytes; and phenotype variability or changes in adipocytes in PVAT have been suggested to lead to different pathological roles in vascular disorders [1]. To this point, interestingly, researchers reported that decreased PVAT ‘browning’ in thoracic aortas of aging spontaneously hypertensive rats (8 wks vs. 16 wks) was associated with attenuated vasodilation effects of PVAT [43], but ATRAP in brown adipose tissue does not affect the development of obesity-related metabolic disorders [44]. Future studies investigating the relationship between changes in adipocytes phenotype and PVAT effects would provide significant mechanistic insight in the SHRSP.ZF model. Additionally, we need to investigate where the angiotensin II is produced that activates the AT1 receptor on PVAT. Since the angiotensinogen mRNA level was decreased in PVAT of SHRSP.ZF (Figure 2C), angiotensin II production in PVAT might be decreased in SHRSP.ZF, despite observations of increased angiotensin II levels in arteries [7]. Enhanced stimulation of AT1 receptor by angiotensin II released from other tissues, including arteries, may have negatively regulated angiotensinogen level in PVAT. Furthermore, the relationship between AT1 receptor activity of PVAT and production/release of apelin and the NAC-sensitive factor, which mediate the enhancing vasodilatory function of PVAT, warrants further study.

In conclusion, we found that exogenous incubation of arteries with apelin could reproduce the PVAT-enhanced relaxation effect of adipose tissue, and that another PVAT factor, its identity undetermined, was sensitive to NAC. These factors could be responsible for mediating the compensatory vasodilatory role of PVAT under conditions of impaired vasodilation observed in mesenteric artery of SHRSP.ZF with MetS. Furthermore, a decrease in ATRAP, leading to an increase in AT1 receptor activity, in PVAT may initiate the deterioration of the PVAT compensatory functions. Further research to understand PVAT in the SHRSP.ZF model may identify novel therapeutic strategies to regulate AT1 receptor signaling in PVAT and protect against arterial PVAT dysfunction while reducing development of cardiovascular diseases related to MetS.

## 4. Materials and Methods

### 4.1. Experimental Animals

The SHRSP.ZF strain was established by the Disease Model Cooperative Research Association and with WKY were purchased from Japan SLC, Inc. (Hamamatsu, Shzuoka, Japan) then housed in groups of 3–animals per cage within an animal care center with a 12-h light-dark cycle at constant room temperature of 21–23 °C with a humidity of 35–65%. Rats were provided a standard chow (CE-2; Clea Japan Inc., Meguro-ku, Tokyo, Japan) and water ad libitum during the experimental period. Experiments described below were started after a habituation period of one week. All procedures were performed in accordance with the guidelines for the Care and Use of Laboratory Animals at Mukogawa Women’s University (P-12-2015-02-A, P-12-2016-02-A; April, 2015 and 2016).

In the first experiments, to assess vascular reactivity changes in increasing age, male SHRSP.ZF assigned to one of three groups according to their age (weeks) at the time of in vitro vasodilation studies: group 1, 17 wks (*n* = 10); groups 2, 20 wks (*n* = 10 each); and group 3, 23 wks (*n* = 10). In the second experiments, to assess effects of adipokines and blockers on vasodilation, male SHRSP.ZF at 20 wks were used (*n* = 25). mRNA and protein expressions were compared between SHRSP.ZF and normal healthy WKY at 20 wks (*n* = 5). This time-point was selected based on the first series of experiments investigating the changes in modulation of vascular tonus by PVAT in SHRSP.ZF.

### 4.2. Chemicals

The chemicals used in the present study are as follows: acetylcholine chloride (Daiichi Pharmaceutical Co., Ltd., Chuo-ku, Tokyo, Japan), l-phenylephrine hydrochloride and NAC (Sigma-Aldrich Co., LLC., St. Louis, MO, USA), vaspin (mouse) and visfatin (rat) (cat. no. AG-40A-0094 and AG-40A-0058, Adipogen Corp., San Diego, CA, USA), omentin (rat) (cat. no. ALX-522-132, Enzo Life Sci., Inc., Farmingdale, NY, USA), and apelin-12 (human, rat, mouse, bovine) (cat. no. 057-23, Phoenix Pharmaceuticals, Inc., Burlingame, CA, USA). Other chemicals of analytical reagent grade were purchased from Nacalai Tesque Inc. (Kyoto, Kyoto, Japan). Stock solutions of test-compounds were prepared in distilled water.

### 4.3. Determination of Metabolic Parameters

Systolic blood pressure was measured by a tail-cuff method using a blood pressure monitor (Model MK-2000, Muromachi Kikai Co., Chuo-ku, Tokyo, Japan) one week before the experimental date of vasodilation studies, as described previously [45]. The systolic blood pressure values were derived from an average of at least five measurements per animal at each time point.

Before collecting tissues from animals for experiments, body weight, body length and waist length were measured under anesthesia (sodium pentobarbital, 80 mg/kg, i.p.), and then blood was drawn from the abdominal aortas of non-fasting rats. The serum was separated by centrifugation at 1000× *g* for 10 min at 4 °C. Serum levels of triglyceride, glucose, insulin, and thiobarbituric acid reactive substances (TBARS), as an index of oxidative stress, were determined using commercial kits; the triglyceride E-test (Wako Pure Chemical Ind. Ltd., Osaka, Osaka, Japan), glucose CII-test (Wako Pure Chemical), the rat insulin detection kit (Morinaga Biochemistry Lab., Yokohama, Kanagawa, Japan), and TBARS Assay Kit (Cayman Chemical Co., Ann Arbor, MI, USA).

### 4.4. Determination of Vasodilation

Superior mesenteric arteries with PVAT were taken from each rat and prepared for myograph experiments as previously described [4]. Briefly, from each rat the superior mesenteric artery was cleaned or not cleaned of surrounding PVAT cut into approximately 3-mm wide rings and mounted isometrically at an optimal resting tension of 0.3 g in a 10 mL organ baths filled with a Krebs–Henseleit solution (pH 7.4; in mM: NaCl 118.4, KCl 4.7, MgSO_4_ 1.2, CaCl_2_ 2.5, NaHCO_3_ 25, KH_2_PO_4_ 1.2, and glucose 11.1) bubbled with 95% O_2_–5% CO_2_ at 37 °C.

For each ring preparation with or without PVAT [PVAT(+) or PVAT(−)], a stable level of isometric contraction was established by addition of phenylephrine (0.1–0.3 µM) before measuring relaxations following the addition of increasing cumulative concentrations (final bath concentrations) of endothelium-dependent vasodilator, acetylcholine (0.1 nM–1 μM), or a NO donor, sodium nitroprusside (0.1 nM–1 μM). In some ring preparations without PVAT [PVAT(−)], after a stable contraction was obtained by adding phenylephrine, relaxation was elicited using acetylcholine in the presence or absence of adipokines, such as omentin (30 ng/mL, 20 min), apelin (100 ng/mL, 20 min), vaspin (3 ng/mL, 20 min), or visfatin (30 ng/mL, 20 min). The treatment conditions of these adipokines were taken from previous studies by another group [15,21,34,46]. Furthermore, in some ring preparations with PVAT [PVAT(+)], after a stable contraction was obtained, relaxation was elicited using acetylcholine in the presence or absence of NAC (4 mM, 15 min). This NAC-treatment protocol was based on previous studies by other researchers [27,36].

Isometric tension changes were measured with a force-displacement transducer (Model t-7; NEC San-Ei, Toyoshima-ku, Tokyo, Japan) coupled to a dual channel chart recorder (Model 8K21; NEC San-Ei). Individual concentration-response curves were analyzed by nonlinear regression curve fitting of relaxation-drug concentration relationships to determine the negative log EC_50_ and E_max_ using GraphPad Prism^®^ (ver. 5.0, San Diego, CA, USA).

### 4.5. Determination of mRNA and Protein Expression

PVAT samples were isolated from mesenteric artery of SHRSP.ZF and WKY, and frozen in liquid nitrogen and stored in a deep freezer (−70 °C) until RNA or protein extraction.

Quantitative real-time PCR assays were performed to examine mRNA transcript levels of angiotensinogen, AT1 receptor, ATRAP, and apelin as previously described [47]. Briefly, total RNA was extracted from tissues and purified using the RNeasy fibrous tissue kit, according to the manufacturer’s instructions (Qiagen, Mississauga, Ontario, Canada); average yield was 100 ng. Real-time measurements of target gene expression were carried out using TaqMan RNA-to-CT 1-step kit and a LightCycler 1.5 (Roche Diagnostics Japan K.K., Minato-ku, Tokyo, Japan). Commercially available gene-specific probes were used (Roche Applied Science, Universal ProbeLibrary product ID: angiotensinogen, 04685059001; AT1 receptor, 04688503001; ATRAP, 04684982001; apelin, 04686896001; ribosomal protein 18S, 04688937001; β-glucronidase, 04688015001; β-actin, 04686900001), and gene-specific primers designed by Assay Design Center (Roche Molecular Systems, Inc., Pleasanton, CA, USA) were purchased from Life Technologies Japan, Ltd. (Minato-ku, Tokyo, Japan). A triad housekeeping gene expression approach (ribosomal protein 18S, β-glucronidase, β-actin) was used for normalization of sample material and the efficiencies for primer sets for each were included in all calculations. The amount of target gene was normalized to the reference gene to obtain the relative threshold cycle (ΔC_T_), and then related to the C_T_ of the level in WKY at 20 wks or SHRSP.ZF at 20 wks to obtain the relative expression level (2^−ΔΔCT^) of target gene. The results were taken from three independent experimental runs.

Western blot assays were performed to examine protein expressions of omentin, apelin, visfatin and vaspin, as previously described [48]. Proteins (40–50 μg) in PVAT were separated by SDS-PAGE and transferred to membranes. Specific proteins were detected using antibodies for either omentin (cat. no. MAB8074, 1:2500 dilution, R&D systems, Inc., Minneapolis, MN, USA), apelin (cat. no. ab125213, 1:2500 dilution, Abcam, Inc., Cambridge, MA, USA), visfatin (cat. no. ab45890, 1:5000 dilution, Abcam, Inc.), vaspin (cat. no. sc-79815, 1:5000 dilution, Santa Cruz Biotechnology, Inc., Dallas, TX, USA), or β-actin (internal control; cat. No. A5316, 1:5000 dilution, Sigma-Aldrich Co., LLC., St. Louis, MO, USA). Semi-quantitative analysis of immunoreactive bands’ intensities was conducted using Image J ver. 10.2 (National Institute of Health, Bethesda, MD, USA). Protein data are expressed as the ratio of the immunoreactive band intensities for each adipokine relative to β-actin.

### 4.6. Data Analysis

Data are expressed as mean ± SEM. *n* = number of animals. Statistical comparisons of means between groups were performed using Student’s *t*-test or one-way ANOVA followed by Bonferroni post-hoc test. Differences were considered significant at *p* < 0.05.

## Figures and Tables

**Figure 1 ijms-20-00106-f001:**
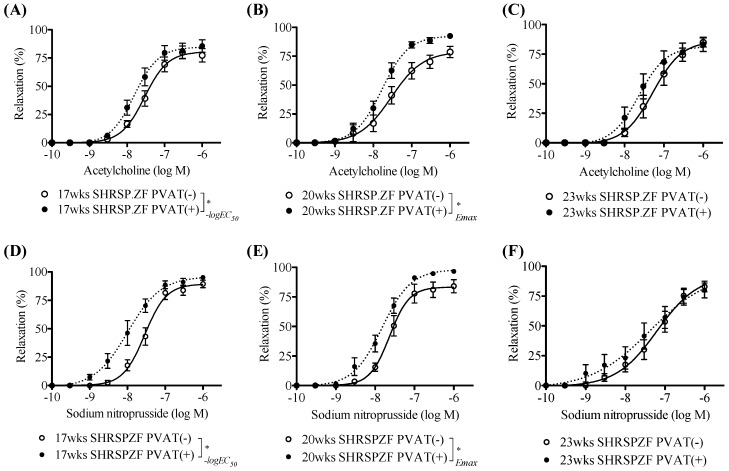
Vasodilation in response to acetylcholine (**A**–**C**) and sodium nitroprusside (**D**–**F**) in superior mesenteric arteries with (+) or without (−) perivascular adipose tissues (PVAT) from SHRSP.Z-*Lepr^fa^*/IzmDmcr rats (SHRSP.ZF) at 17, 20, and 23 weeks of age (wks). * *p* < 0.05.

**Figure 2 ijms-20-00106-f002:**
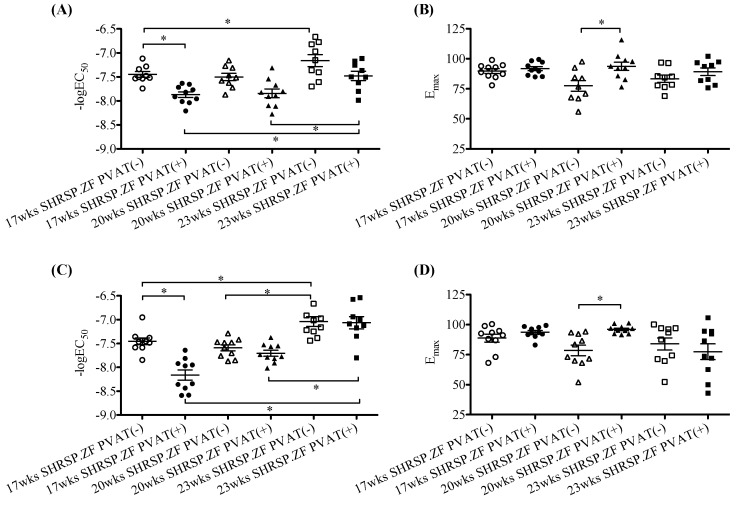
Negative log EC_50_ and E_max_ of relaxations in response to acetylcholine (**A**,**B**) and sodium nitroprusside (**C**,**D**) in isolated superior mesenteric arteries with (+) or without (−) perivascular adipose tissue (PVAT) from SHRSP.Z-*Lepr^fa^*/IzmDmcr rats (SHRSP.ZF) at 17, 20, and 23 weeks of age (wks). * *p* < 0.05.

**Figure 3 ijms-20-00106-f003:**
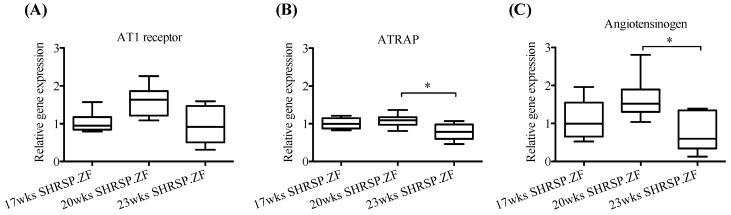
mRNA expression of angiotensin II type 1 (AT1) receptor (**A**), AT1 receptor-associated protein (ATRAP, (**B**)), and angiotensinogen (**C**) in superior mesenteric arterial perivascular adipose tissues (PVAT) of SHRSP.Z-*Lepr^fa^*/IzmDmcr rats (SHRSP.ZF) at 17, 20, and 23 weeks of age (wks). * *p* < 0.05.

**Figure 4 ijms-20-00106-f004:**
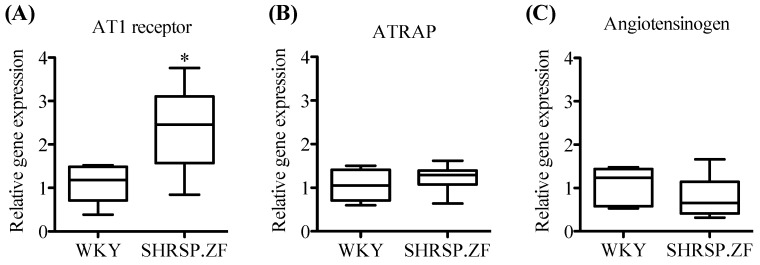
mRNA expression of angiotensin II type 1 (AT1) receptor (**A**), AT1 receptor-associated protein (ATRAP, **B**), and angiotensinogen (**C**) in superior mesenteric arterial perivascular adipose tissues (PVAT) of SHRSP.Z-*Lepr^fa^*/IzmDmcr rats (SHRSP.ZF) and their normal control Wistar-Kyoto rats (WKY) at 20 weeks of age. * *p* < 0.05.

**Figure 5 ijms-20-00106-f005:**
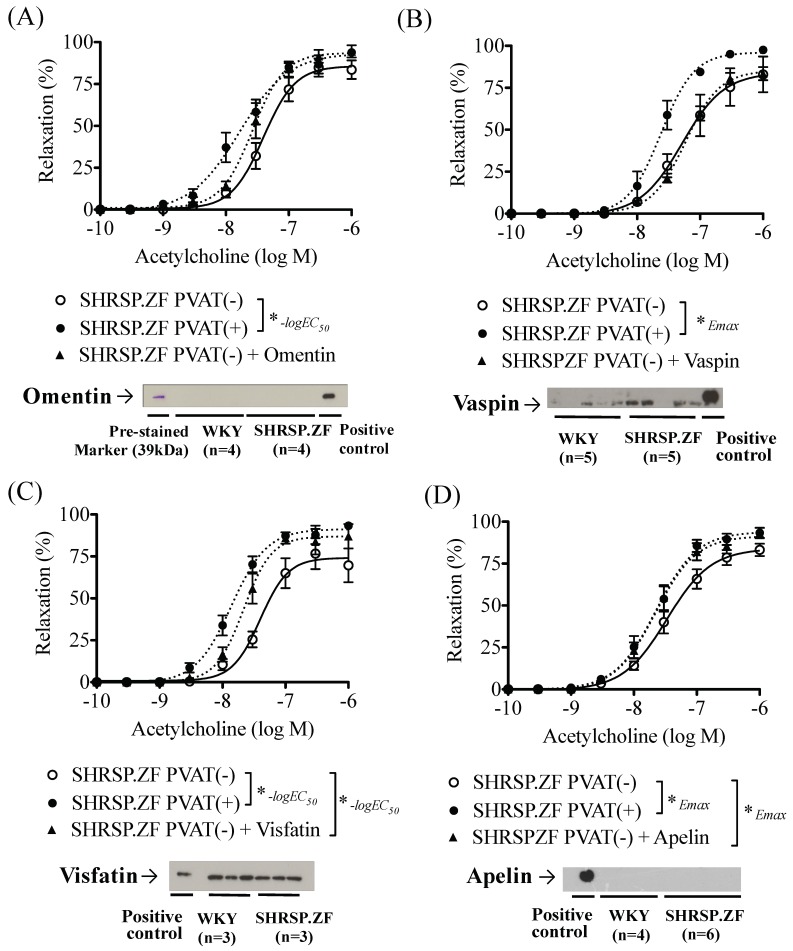
Effects of adipokines ((**A**), omentin; (**B**), vaspin, (**C**); visfatin, (**D**); apelin) on vasodilation in response to acetylcholine in superior mesenteric arteries with (+) or without (−) perivascular adipose tissues (PVAT) and its protein expression in PVAT of SHRSP.Z-*Lepr^fa^*/IzmDmcr rats (SHRSP.ZF) at 20 weeks of age. Upper panel show changes in the relaxations. Lower panel show representative data indicating immunoreactive bands corresponding to each adipokines for SHRSP.ZF and their normal control Wistar-Kyoto rats (WKY). * *p* < 0.05.

**Figure 6 ijms-20-00106-f006:**
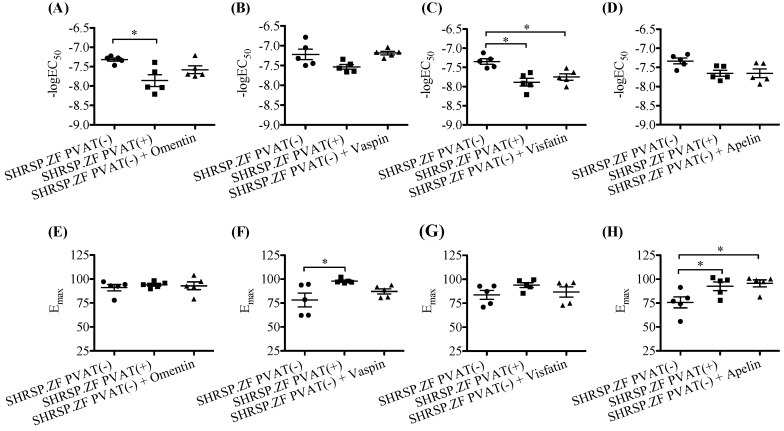
Negative log EC_50_ and E_max_ of relaxations in response to acetylcholine in the presence or absence of adipokines ((**A**,**E**), omentin; (**B**,**F**), vaspin, (**C**,**G**); visfatin, (**D**,**H**); apelin) in isolated superior mesenteric arteries with (+) or without (−) perivascular adipose tissue (PVAT) from SHRSP.Z-*Lepr^fa^*/IzmDmcr rats (SHRSP.ZF) at 20 weeks of age. * *p* < 0.05.

**Figure 7 ijms-20-00106-f007:**
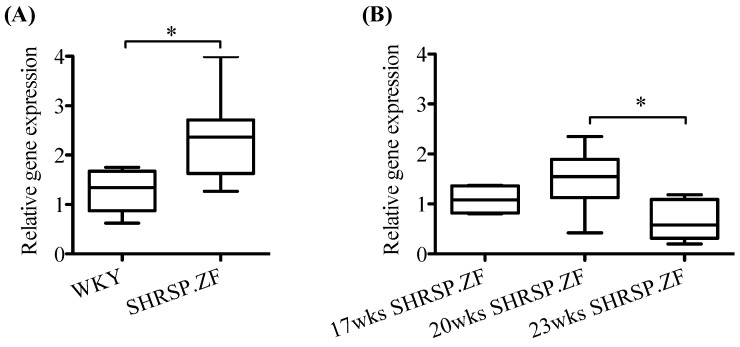
mRNA expression of apelin in superior mesenteric arterial perivascular adipose tissues of SHRSP.Z-*Lepr^fa^*/IzmDmcr rats (SHRSP.ZF) and their normal control Wistar-Kyoto rats (WKY) at 20 weeks of age (wks) (**A**), and those of SHRSP.ZF at 17, 20, and 23 wks (**B**). * *p* < 0.05.

**Figure 8 ijms-20-00106-f008:**
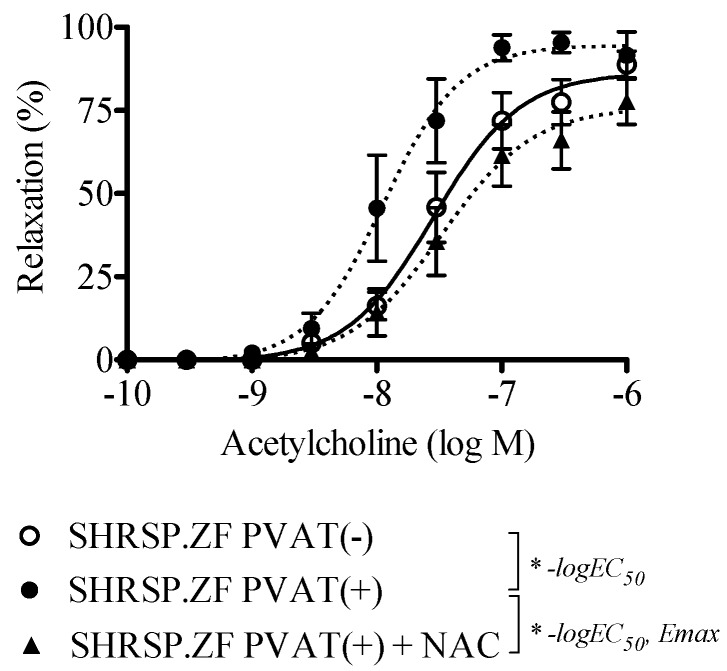
Effects of *N*-acetyl-l-cysteine (NAC; 4 mM) on vasodilation in response to acetylcholine in superior mesenteric arteries with (+) or without (−) perivascular adipose tissues (PVAT) from SHRSP.Z-*Lepr^fa^*/IzmDmcr rats (SHRSP.ZF) at 20 weeks of age. * *p* < 0.05.

**Figure 9 ijms-20-00106-f009:**
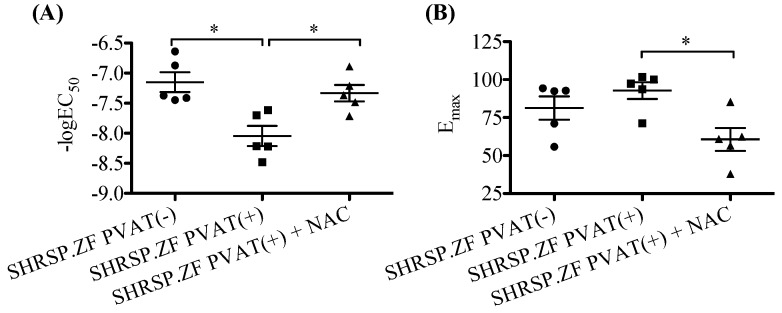
Negative log EC_50_ (**A**) and E_max_ (**B**) of relaxations in response to acetylcholine in the presence or absence of *N*-acetyl-l-cysteine (NAC; 4 mM) in isolated superior mesenteric arteries with perivascular adipose tissue [PVAT(+)] from SHRSP.Z-*Lepr^fa^*/IzmDmcr rats (SHRSP.ZF) at 20 weeks of age. * *p* < 0.05.

**Table 1 ijms-20-00106-t001:** Changes in metabolic parameters in SHRSP.Z-*Lepr^fa^*/IzmDmcr rats (SHRSP.ZF) at 17, 20, and 23 weeks of age (wks).

	SHRSP.ZF		
	17 wks	20 wks	23 wks
Body weight (g)	414 ± 5	438 ± 7	462 ± 10 *
Waist length/body length (cm/cm)	1.03 ± 0.01	1.00 ± 0.01	1.03 ± 0.01
Systolic blood pressure (mmHg)	215 ± 7	205 ± 8	234 ± 9 ^#^
Serum triglycerides (mg/100 mL)	749 ± 35	558 ± 71	709 ± 68
Serum glucose (mg/100 mL)	386 ± 24	328 ± 29	360 ± 24
Serum insulin (ng/mL)	49.8 ± 5.5	41.3 ± 5.2	63.2 ± 5.2 ^#^
Serum TBARS (µM)	14.7 ± 0.9	13.1 ± 0.9	19.9 ± 1.8 *^, #^

TBARS: thiobarbituric acid reactive substances. Results are expressed as the mean ± SEM. * *p* < 0.05, as compared with SHRSP.ZF at 17 wks, ^#^
*p* < 0.05, as compared with SHRSP.ZF at 20 wks. *n* = 10 at each age.

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
