# Peer review of "Perivascular Adipose Tissue-Enhanced Vasodilation in Metabolic Syndrome Rats by Apelin and N-Acetyl–l-Cysteine-Sensitive Factor(s)"

_ijms, 2018, doi:10.3390/ijms20010106_

Reviewer 1 Report

The manuscript by Kagota et al. investigated the mechanisms of age-related changes and responsible factor involved in enhancing effects of mesenteric arterial PVAT in SHRSP.ZF. Although the results showed the disappearance of PVAT-mediated regulation of vasorelaxation in response to acetylcholine coincided with a decreased ATRAP mRNA level in adipose tissue of superior mesenteric arteries of SHRSP.ZF with MetS, it is unclear that the decrease in ATRAP mRNA is related to the enhancing effects of PVAT on vascular relaxations. On the other hand, apelin mRNA in mesenteric PVAT was higher in SHRSP.ZF than controls and the incubation of tissues with exogenous apelin resulted in enhanced acetylcholine induced relaxations. It is interesting however the protein expressions in mesenteric PVAT were no detectable and CT value is not written. Treatment of tissues with NAC reduced relaxations in mesenteric artery with PVAT but there is no evidence that HNO improves the relaxation in this manuscript. These findings of the study seem to be significant and interesting to the readers. However, additional explanation or/and experiments are needed.

 Comment 1: The authors found interesting results for acetylcholine-induced relaxation in SHRSP.ZF with/without PVAT. How was sodium nitroprusside-induced endothelial-independent relaxation? Can you describe the data on text and/or figure?

 Comment 2: The authors carefully showed the vasodilation data as table 2, 3 and 4. For the readers, would you please show EC50 and Emax using bar graph on each figure?  

 Comment 3: ATRAP in 20 wks SHRSP.ZF was significantly lower than 23 wks in Figure 2 and the authors suggested that AT1 receptor activity in PVAT is related to the deterioration of the enhancing effects of PVAT on vascular relaxations. Can you block ATRAP activity using siRNA or inhibitor and check the vasodilation? Or did you check it using AT1R?  

 Comment 4: mRNA expression of apelin was significantly increased in 20 wks SHRSP.ZF in Figure 3. How was the expression in 23 wks? Following your conclusion, mRNA expression of apelin is decreased in 23 wks SHRSP.ZF compared to age-matched WKY or 20 wks SHRSP.ZF, isn’t it? And can you show CT values since the protein expression of apelin was no detectable shown in Figure 4.

 Comment 5: In Table 1, serum triglycerides in 20wks was lower than others. Was there no significance? Is the value correct?

 Comment 6: In table 2, was there any difference between SHRSP.ZF without PVAT (-) of 17, 20 and 23 wks?

Author Response

To Reviewer 1

We would like to thank the reviewer for your appraisals of the manuscript and insightful comments. Our responses (bolded) to each of the reviewer’s comments (italicized) are provided below.

The manuscript by Kagota et al. investigated the mechanisms of age-related changes and responsible factor involved in enhancing effects of mesenteric arterial PVAT in SHRSP.ZF. Although the results showed the disappearance of PVAT-mediated regulation of vasorelaxation in response to acetylcholine coincided with a decreased ATRAP mRNA level in adipose tissue of superior mesenteric arteries of SHRSP.ZF with MetS, it is unclear that the decrease in ATRAP mRNA is related to the enhancing effects of PVAT on vascular relaxations. On the other hand, apelin mRNA in mesenteric PVAT was higher in SHRSP.ZF than controls and the incubation of tissues with exogenous apelin resulted in enhanced acetylcholine induced relaxations. It is interesting however the protein expressions in mesenteric PVAT were no detectable and CT value is not written. Treatment of tissues with NAC reduced relaxations in mesenteric artery with PVAT but there is no evidence that HNO improves the relaxation in this manuscript. These findings of the study seem to be significant and interesting to the readers. However, additional explanation or/and experiments are needed.

Comment 1: The authors found interesting results for acetylcholine-induced relaxation in SHRSP.ZF with/without PVAT. How was sodium nitroprusside-induced endothelial-independent relaxation? Can you describe the data on text and/or figure? 

We found that sodium nitroprusside-induced relaxations were affected by PVAT in the same way as acetylcholine. To address Reviewer 1’s comment, we have added these data to the Results section (Fig. 1 and Table 1), and provided text descriptions in the Abstract (lines 19-22), Results (lines 99-132), and Discussion (lines 273-275).

 Comment 2: The authors carefully showed the vasodilation data as table 2, 3 and 4. For the readers, would you please show EC50 and Emax using bar graph on each figure?  

To address Reviewer 1’s comment, we have showed these data using scatter plot graphs (Figs. 2, 6 and 9).

 Comment 3: ATRAP in 20 wks SHRSP.ZF was significantly lower than 23 wks in Figure 2 and the authors suggested that AT1 receptor activity in PVAT is related to the deterioration of the enhancing effects of PVAT on vascular relaxations. Can you block ATRAP activity using siRNA or inhibitor and check the vasodilation? Or did you check it using AT1R?  

We would like to thank Reviewer 1 for providing this helpful suggestion to better understand the underlying mechanism. Unfortunately, these tools to pursue inhibition of ATRAP have yet to be applied in tissue and doing so would require extensive experimental design considerations, for example, to employ organ culture conditions. However, we can share with the reviewers that we are pursuing the question of whether administration in vivoof an AT1 receptor antagonist treatment restores the enhancing effect of PVAT on vasodilation with increasing age in MetS rats because our working hypothesis is that continuous AT1 activation in PVAT induces deterioration of its function. And, as alluded to by the reviewer’s suggestion, there is the possibility that AT1 activation directly inhibits the production/release of the relaxing factor(s) from PVAT. To investigate this latter possibility, in parallel to the in-vivo study, we are testing whether in vitrotreatment with AT1 receptor antagonist affects the relaxation in isolated mesenteric artery with PVAT in SHRSP.ZF. This large study on the effects of an AT1 receptor antagonist treatment on vasodilation in MetS rats is not yet completed, but we expect the in vitroand in vivodata will be reported together. 

To address the Reviewer’s comments, we have added text regarding the relationship between AT1 receptor activity of PVAT and production/release of the factor(s), which mediates the enhancing vasodilatory function of PVAT, to the Discussion (lines 372-375).

 Comment 4: mRNA expression of apelin was significantly increased in 20 wks SHRSP.ZF in Figure 3. How was the expression in 23 wks? Following your conclusion, mRNA expression of apelin is decreased in 23 wks SHRSP.ZF compared to age-matched WKY or 20 wks SHRSP.ZF, isn’t it? And can you show CT values since the protein expression of apelin was no detectable shown in Figure 4. 

We would like to thank Reviewer 1 for drawing our attention to these points. Yes, we investigated whether apelin mRNA expression in PVAT decreased with aging in SHRSP.ZF. We have added these data to Fig. 7B, and described the results in text of the Abstract (line 27), Results (lines 202-203), and Discussion (lines 287-288, and 323-326) sections. Please note that to incorporate this result into the revised MS, we restructured the presentation of figures in the manuscript. Namely, Fig. 3D (WKY vs. SHRSP.ZF) in the original MS was moved to Fig. 7, and is exhibited as 7A with new data exhibited as 7B in the revised MS. 

Generally, the acceptable range of CT values for quantitative RT-PCR methods is considered to be 17 to 32. In the present study, the range of CT values for apelin in PVAT was 23.24-25.75. As requested by the reviewer, we have added this information to the results section (line 204).

 Comment 5: In Table 1, serum triglycerides in 20wks was lower than others. Was there no significance? Is the value correct? 

Yes, we have double checked the data and confirm that the correct values and statistics are reported in Table 1. 

 Comment 6: In table 2, was there any difference between SHRSP.ZF without PVAT (-) of 17, 20 and 23 wks?

These relaxations in response to either acetylcholine or sodium nitroprusside were decreased with aging in SHRSP.ZF. In Fig. 2, the symbol * is added to indicate the statistical significance for readers.

Reviewer 2 Report

This is an interesting study by Kagota et al. examining potential mechanism(s) underlying the important of PVAT-enhancing vasodilation in mesenteric artery of SHRSP.ZF with MetS. The authors showed that vascular relaxation in ACh was significantly regulated by PVAT in SHRSP.ZF. They concluded that the increase in AT1 receptor by HNO and apelin in PVAT contributes to the age-dependent deterioration of vasodilation enhancing effects of mesenteric artery in SHRSP.ZF. While this study well performed, there are several items that require additional attention.

Table 1 suggested that metabolic parameters became dramatically worse between 20 and 23 weeks. Did the authors examine whether PVAT-mediated vascular relaxation response was also enhanced in SHRSP. ZF. at 23 weeks? Then the authors should discuss whether these data are an aging process in SHRSP.ZF.

The authors concluded that the PVAT-mediated vascular relaxation was due to H2S, not NO. The authors should measure the production of H2S and NO and show H2S is indeed increased in SHRSP.ZF.

The expression levels of AT1 receptor were increased in SHRSP.ZF PVAT? Did the authors examine whether circulating Ang II was also elevated in SHRSP.ZF? The authors should discuss about the relationship among Ang II, AT1 receptor and ATRAP in SHRSP.ZF.

In the lower panels of Figure 4, what was used for normalization of sample? 

Each protein levels of adipokines was shown in Results (the account for the Figure 4). The way of calculation is unclear.

Author Response

To Reviewer 2

We would like to thank the reviewers for their appraisals of the manuscript and insightful comments. Our responses (bolded) to each of the reviewers’ comments (italicized) are provided below.

This is an interesting study by Kagota et al. examining potential mechanism(s) underlying the important of PVAT-enhancing vasodilation in mesenteric artery of SHRSP.ZF with MetS. The authors showed that vascular relaxation in ACh was significantly regulated by PVAT in SHRSP.ZF. They concluded that the increase in AT1 receptor by HNO and apelin in PVAT contributes to the age-dependent deterioration of vasodilation enhancing effects of mesenteric artery in SHRSP.ZF. While this study well performed, there are several items that require additional attention.

Table 1 suggested that metabolic parameters became dramatically worse between 20 and 23 weeks. Did the authors examine whether PVAT-mediated vascular relaxation response was also enhanced in SHRSP.ZF at 23 weeks? Then the authors should discuss whether these data are an aging process in SHRSP.ZF.

We regret that our results were not more clear for the reader. The PVAT-mediated vascular response disappears at 23 wks. Our previous work had found that the effect was absent at 30 weeks so our present study has refined the timeline for this deterioration of PVAT function. To clarify the time-course and ageing effect, we have added text to the Discussion (lines 267-279). to draw attention to the development of impaired PVAT functions in the aging process of SHRSP.ZF as noted by Reviewer 2in 

 The authors concluded that the PVAT-mediated vascular relaxation was due to H2S, not NO. The authors should measure the production of H2S and NO and show H2S is indeed increased in SHRSP.ZF.

We understand the Reviewer’s point and thank them for these suggested experiments. As we mentioned in the Introduction (lines 84-85), the present study is the start to our investigation of adipose-derived factors. The results of our current study provide a rationale to pursue direct measurements of H2S production by PVAT and the regulation of its enzymatic synthesis. In the revised manuscript we have acknowledged the limitations to the scope of the current study in the Discussion (lines 361-362) and revised our conclusions (lines, 28, 290, and 375-377) and the title of the manuscript accordingly.

 The expression levels of AT1 receptor were increased in SHRSP.ZF PVAT? Did the authors examine whether circulating Aag II was also elevated in SHRSP.ZF? The authors should discuss about the relationship among Ang II, AT1 receptor and ATRAP in SHRSP.ZF.

Fig. 4 shows the mRNA level of AT1 receptor was increased in PVAT of SHRSP.ZF at 20 wks compared to age-matched normal control rats (WKY). In Figure 3, the level of AT1 mRNA level was unchanged by age in SHRSP.ZF, which we have interpreted to mean that AT1 receptor is continuously higher in MetS PVAT than in control group PVAT (WKY). Combining these findings with the ATRAP expression data, we suggest that “… activation of AT1 receptor in PVAT initiates the deterioration of the PVAT’s modification of vascular tone in MetS then declining ATRAP sustains the dysfunction” in the Discussion (lines 301-302). To clarify our point, we made some changes to the text (lines 297-299).

We did not measure circulating angiotensin II level i.e. its serum level, in SHRSP.ZF. However, systemic levels may not directly reflect what is happening at the interface of PVAT and arteries. It is recognized that localized tissue renin-angiotensin system plays a critical role in the altered vasodilation in MetS. This is supported by evidence of increased angiotensin II levels in intact arteries of SHRSP.ZF. On the other hand, in the case of PVAT, angiotensinogen mRNA level was decreased in PVAT of SHRSP.ZF with aging (Fig. 3). Together this evidence raises the possibility that enhanced activation of PVAT AT1 receptor negatively regulates expression of angiotensinogen. In other words, angiotensin II in the local PVAT environment may actually decrease in SHRSP.ZF. We have revised the text to clarify these discussion points in the Discussion (lines 372-375).

 In the lower panels of Figure 4, what was used for normalization of sample? Each protein levels of adipokines was shown in Results (the account for the Figure 4). The way of calculation is unclear.

We regret the details for normalization were omitted from the original submission and the calculation of protein levels was not clear for readers. In Western blot experiments, beta-actin was used for normalization of samples. This information is included in the Methods section (line 485). The description of calculations used for analyses of protein data has been revised in the Methods (lines 487-488), and the data are now also included in the Results (lines 195-199).

Round  2

Reviewer 1 Report

General Comments:

The authors seem to address the reviewer’s comments. However, the lines which authors indicated in the cover letter don’t fit in the manuscript. It is difficult to find the author’s changes.

 Concerns;

1. Please check your replies on Review’s 1-comment 3 and 4. For example, Author’s reply for comment 3 was “To address the Reviewer’s comments, we have added text regarding the relationship between AT1 receptor activity of PVAT and production/release of the factor(s), which mediates the enhancing vasodilatory function of PVAT, to the Discussion (lines 372-375).” However, line 372-375 is not discussion part, but Materials and Methods. Please fix them.

 2. In new Figure 2, 6, 9, can you narrow ranges of y axis? For example, y axis might be -9 to -6.5 in new Figure 2A. I think authors should emphasize these data.  

Author Response

Response to the comments from the 2nd round

Concerns 1: Please check your replies on Reviewer’s 1-comment 3 and 4. For example, Author’s reply for comment 3 was “To address the Reviewer’s comments, we have added text regarding the relationship between AT1 receptor activity of PVAT and production/release of the factor(s), which mediated the enhancing vasodilatory function of PVAT, to the Discussion (lines 372-375). However, line 372-375 is not discussion part, but Materials and Methods. Please fix them.

We apologize for these errors. The correct line numbers have been inserted in our replies to reviewers’ comments below (underlined). 

Concerns 2: In new Figures 2, 6, 9, can you narrow ranges of y axis? For example, y axis might be -9 to -6.5 in new Figure 2A. I think authors should emphasize these data.

The ranges for the y-axis of Fig. 2, 6 and 9 are re-formatted according to this suggestion.

 Response to the comments from the 1st round

We would like to thank the reviewer for your appraisals of the manuscript and insightful comments. Our responses (bolded) to each of the reviewer’s comments (italicized) are provided below.

Comment 1: The authors found interesting results for acetylcholine-induced relaxation in SHRSP.ZF with/without PVAT. How was sodium nitroprusside-induced endothelial-independent relaxation? Can you describe the data on text and/or figure? 

We found that sodium nitroprusside-induced relaxations were affected by PVAT in the same way as acetylcholine. To address Reviewer 1’s comment, we have added these data to the Results section (Figs. 1 and 2), and provided text descriptions in the Abstract (lines 19-22), Results (lines 93-101), and Discussion (lines 185-187).

 Comment 2: The authors carefully showed the vasodilation data as table 2, 3 and 4. For the readers, would you please show EC50 and Emax using bar graph on each figure?  

To address Reviewer 1’s comment, we have showed these data using scatter plot graphs (Figs. 2, 6 and 9). Also, we have changed the ranges of y axis of these Figs, according to the Editor 1 suggestion.

 Comment 3: ATRAP in 20 wks SHRSP.ZF was significantly lower than 23 wks in Figure 2 and the authors suggested that AT1 receptor activity in PVAT is related to the deterioration of the enhancing effects of PVAT on vascular relaxations. Can you block ATRAP activity using siRNA or inhibitor and check the vasodilation? Or did you check it using AT1R?  

We would like to thank Reviewer 1 for providing this helpful suggestion to better understand the underlying mechanism. Unfortunately, these tools to pursue inhibition of ATRAP have yet to be applied in tissue and doing so would require extensive experimental design considerations, for example, to employ organ culture conditions. However, we can share with the reviewers that we are pursuing the question of whether administration in vivoof an AT1 receptor antagonist treatment restores the enhancing effect of PVAT on vasodilation with increasing age in MetS rats because our working hypothesis is that continuous AT1 activation in PVAT induces deterioration of its function. And, as alluded to by the reviewer’s suggestion, there is the possibility that AT1 activation directly inhibits the production/release of the relaxing factor(s) from PVAT. To investigate this latter possibility, in parallel to the in-vivo study, we are testing whether in vitrotreatment with AT1 receptor antagonist affects the relaxation in isolated mesenteric artery with PVAT in SHRSP.ZF. This large study on the effects of an AT1 receptor antagonist treatment on vasodilation in MetS rats is not yet completed, but we expect the in vitroand in vivodata will be reported together.

To address the Reviewer’s comments, we have added text regarding the relationship between AT1 receptor activity of PVAT and production/release of the factor(s), which mediates the enhancing vasodilatory function of PVAT, to the Discussion (lines 265-267).

 Comment 4: mRNA expression of apelin was significantly increased in 20 wks SHRSP.ZF in Figure 3. How was the expression in 23 wks? Following your conclusion, mRNA expression of apelin is decreased in 23 wks SHRSP.ZF compared to age-matched WKY or 20 wks SHRSP.ZF, isn’t it? And can you show CT values since the protein expression of apelin was no detectable shown in Figure 4. 

We would like to thank Reviewer 1 for drawing our attention to these points. Yes, we investigated whether apelin mRNA expression in PVAT decreased with aging in SHRSP.ZF. We have added these data to Fig. 7B, and described the results in text of the Abstract (line 27), Results (lines 145-147) and Discussion (lines 193-194, and 228-232) sections. Please note that to incorporate this result into the revised MS, we restructured the presentation of figures in the manuscript. Namely, Fig. 3D (WKY vs. SHRSP.ZF) in the original MS was moved to Fig. 7, and is exhibited as 7A with new data exhibited as 7B in the revised MS. 

Generally, the acceptable range of CT values for quantitative RT-PCR methods is considered to be 17 to 32. In the present study, the range of CT values for apelin in PVAT was 23.24-25.75. As requested by the reviewer, we have added this information to the results section (line 147).

Round  3

Reviewer 1 Report

The authors completely address the reviewer’s comments. I do not have any more comments for this. 

Author Response

The authors have proof read the manuscript to make minor corrections for spelling.